# Dynamic 3D Gaussian Tracking for Graph-Based Neural Dynamics Modeling

**Mingtong Zhang**[*1], **Kaifeng Zhang**[*2], **Yunzhu Li**[2]
[1]University of Illinois Urbana-Champaign, [2]Columbia University

**Abstract:** Videos of robots interacting with objects encode rich information about the objects' dynamics. However, existing video prediction approaches typically do not explicitly account for the 3D information from videos, such as robot actions and objects' 3D states, limiting their use in real-world robotic applications. In this work, we introduce a framework to learn object dynamics directly from multi-view RGB videos by explicitly considering the robot's action trajectories and their effects on scene dynamics. We utilize the 3D Gaussian representation of 3D Gaussian Splatting (3DGS) to train a particle-based dynamics model using Graph Neural Networks. This model operates on sparse control particles downsampled from the densely tracked 3D Gaussian reconstructions. By learning the neural dynamics model on offline robot interaction data, our method can predict object motions under varying initial configurations and unseen robot actions. The 3D transformations of Gaussians can be interpolated from the motions of control particles, enabling the rendering of predicted future object states and achieving action-conditioned video prediction. The dynamics model can also be applied to model-based planning frameworks for object manipulation tasks. We conduct experiments on various kinds of deformable materials, including ropes, clothes, and stuffed animals, demonstrating our framework's ability to model complex shapes and dynamics. Our project page is available at https://gs-dynamics.github.io.

**Keywords:** Dynamics Model, 3D Gaussian Splatting, Action-Conditioned Video Prediction, Model-Based Planning

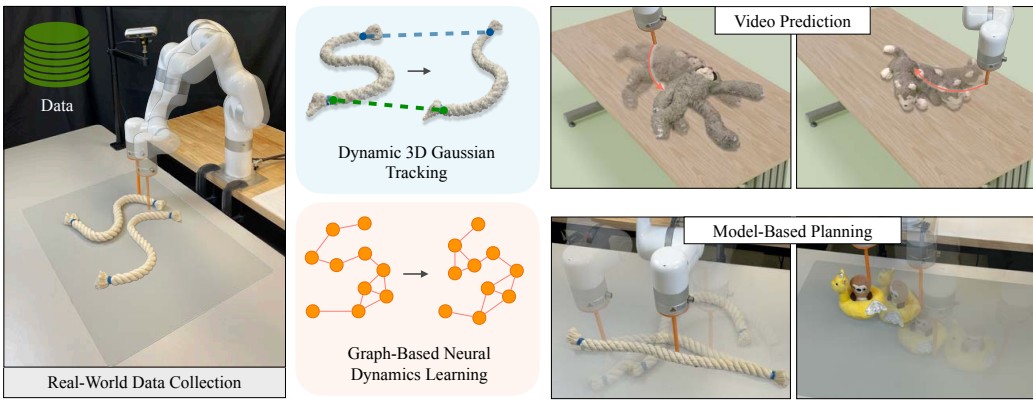

Figure 1: We propose a novel approach for learning a neural dynamics model from real-world data. Using videos captured from robot-object interactions, we obtain dense 3D tracking with a dynamic 3D Gaussian Splatting framework. We train a graph-based neural dynamics model on top of the 3D Gaussian particles for action-conditioned video prediction and model-based planning.

---

[*]Equal contribution.

8th Conference on Robot Learning (CoRL 2024), Munich, Germany.

# 1 Introduction

Humans naturally grasp the dynamics of objects through observation and interaction, allowing them to intuitively predict how objects will move in response to specific actions [1]. This predictive capability enables humans to plan their behavior to achieve specific goals. In robotics, developing predictive models of object motions is critical for model-based planning [2–4]. However, learning such models from real data is challenging due to the complex physics involved in robot-object interactions. Previous works have used simulation environments to learn dynamics models where ground truth 3D object states are available [5–7]. In contrast, real interaction data, such as videos, tend to be difficult to extract 3D information from, thus leading to inefficiencies in model learning.

Recent advancements in 3D Gaussian Splatting have introduced a novel approach to 3D reconstruction. 3D Gaussian Splatting uses a collection of 3D Gaussians with optimizable parameters as object particles. A direct extension of this framework is to optimize the temporal motions of 3D Gaussians, leading to dynamic scene fitting. Nevertheless, such reconstruction methods only fit given dynamic scenes but can not predict object motions into the future.

In this work, we propose a novel method that combines dynamic 3D reconstruction with dynamics modeling. Our method learns a neural dynamics model from real videos for 3D action-conditioned video prediction and model-based planning. To achieve this, we first follow previous dynamic 3D reconstruction approaches [8] to obtain a particle-based representation for dynamic scenes. We extract dense correspondence from long-horizon robot-object interaction videos, which serve as the training data for a dynamics model based on Graph Neural Networks (GNNs). Operating on a spatial graph of control particles, the model predicts object motions under external actions such as robot interactions. To enable dense motion prediction, we design an interpolation scheme to calculate the transformations of 3D Gaussians from sparse control particles, enabling action-conditioned video prediction. The dynamics model can also be incorporated into a model-based planning pipeline, e.g. model predictive control, for object manipulation tasks.

We perform experiments on various objects of deformable materials including ropes, cloths, and toy animals. Results show our method accurately reconstructs 3D Gaussians and maintains coherent correspondences across frames, even with occlusions. The learned dynamics model simulates the physical behaviors of the deformable materials truthfully and generalizes well to unseen actions. Our method outperforms other system identification approaches (such as parameter optimization in physics-based simulators) in motion prediction accuracy and video prediction quality. We also demonstrate the model-based planning performance using our learned model in a range of object manipulation tasks.

# 2 Related Work

**Dense Correspondence from Videos.** Dense correspondence is usually extracted from videos using pixel-wise tracking [9–13]. Such tracking methods are usually formulated as a 2D prediction problem and require large datasets to train. Our approach is more related to another line of work, which focuses on lifting RGB or RGB-D images to 3D and track in the 3D space [14–16]. Recently, Dynamic 3D Gaussian Splatting approaches [8, 17] reconstruct objects as Gaussian particles and optimize per-sequence tracking using rendering loss. Building on this line of work, our method uses 3D point scans of objects as initialization and extracts correspondence for dynamics model training.

A primary incentive for examining point tracking is its potential application to robotics [18, 19]. For instance, RoboTAP [20] demonstrates that pre-trained point tracking models enhance the sample efficiency in visual imitation learning, and ATM [21] shows that predictions of point trajectories provide control guidance for robots. Our method, in contrast, learns neural dynamics on top of particle-based tracking, which generalizes to unseen robot actions and can be naturally integrated with a model-based planning framework.

**Neural Dynamics Modeling of Real-World Objects.** Modeling the dynamics of real-world objects is extremely challenging due to the high complexity of the state space and the variance of physical properties, especially for deformable materials. Using physics-based simulators, previ-

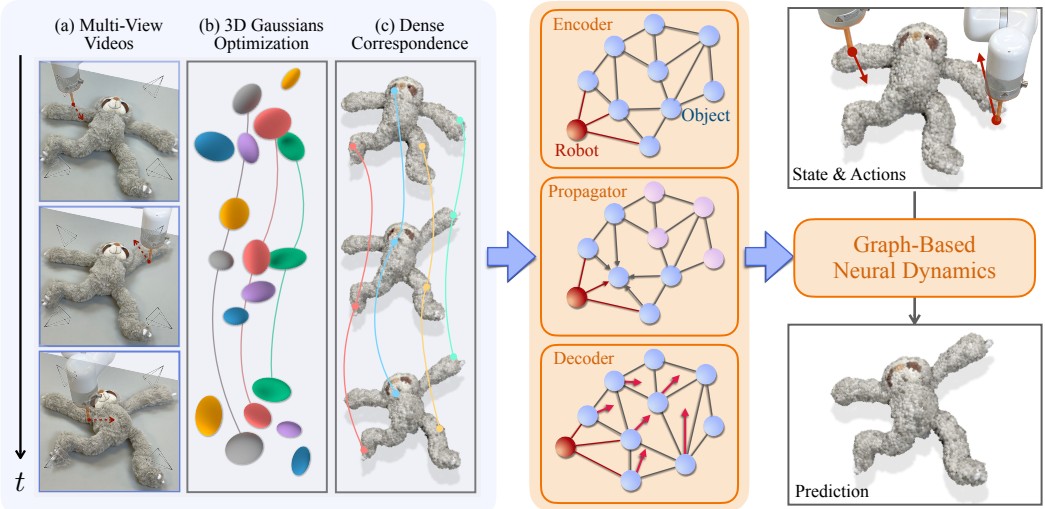

Figure 2: **Overview of Our Framework:** We first achieve dense 3D tracking of long-horizon robot-object interactions using multi-view videos and Dyn3DGS optimization. We then learn the object dynamics through a graph-based neural network. This approach enables applications such as (i) action-conditioned video prediction using linear blend skinning for motion prediction, and (ii) model-based planning for robotics.

ous works have manually specified physical parameters [22], or performed gradient-free parameter search [23] and gradient-based methods [24–26] to optimize low-dimensional physical parameters in a continuous domain. However, these methods require accurate perception models to transfer objects into simulatable assets and thus are limited to constrained, manually designed environments. Neural network-based simulators can also learn object dynamics without requiring the analytical model [27–31]. Especially, graph-based networks can learn the dynamics of various kinds of models such as plasticine [32, 33], cloth [6, 34], fluid [5, 35], etc. Our method builds upon these previous works but reduces the requirement of large-scale offline training data, enabling direct learning from videos, thus potentially eliminating sim-to-real gaps.

**Video Prediction for Robotics.** Autoregressive [36–38] and diffusion models [39–41] trained on Internet-scale data have demonstrated significant capabilities in video generation. Action-conditioned video prediction methods initially showed their capability in predicting future video frames conditioned on actions in simple scenarios, such as atari games [42, 43] and object push [44]. Recently, diffusion-based methods for video prediction [45–49] have created realistic videos of robotic or human manipulations using large-scale data. These models can be applied to generating robot actions by solving robot poses from videos [50] or learning goal-conditioned policies [46, 51]. Compared to these approaches, our model learns action-conditioned video prediction in 3D, thus making it compatible with model-based planning frameworks for various manipulation tasks.

## 3 Method

### 3.1 Preliminary: 3D Gaussian Splatting

3D Gaussian Splatting [52] optimizes a dense set of 3D Gaussians as explicit scene representation. Each Gaussian is defined by a 3D covariance matrix $\Sigma$ centered at $\mu$. The matrix $\Sigma$ decomposes into a rotation matrix $R$ and a scale matrix $S$ by $\Sigma = RSS^T R^T$. For image rendering, 3D Gaussians are projected to 2D using viewing transformation $W$. The covariance matrix in camera coordinates $\Sigma'$ is computed as $\Sigma' = JW\Sigma W^T J^T$, where $J$ is the Jacobian of the affine approximation of the projective transformation.

To optimize the 3D Gaussians, they are projected onto the image plane, where each pixel's color $C$ is determined by a weighted blending:

$$C = \sum_{i \in N} c_i \alpha_i \prod_{j=1}^{i-1} (1 - \alpha_j), \tag{1}$$

where $c_i$ is the color, and $\alpha_i$ is the opacity value of each 3D Gaussian. The loss function is defined between the rendering and the ground truth image, as the $\mathcal{L}_1$ error with a D-SSIM term: $L = (1 - \lambda)\mathcal{L}_1 + \lambda\mathcal{L}_{\text{D-SSIM}}$, where we take $\lambda = 0.2$ in our experiments.

## 3.2 Dynamic 3D Gaussian Splatting with Dense Tracking

3D Gaussians have been proven effective in modeling continuously changing dynamic scenes. For instance, Dynamic 3D Gaussians (Dyn3DGS) [8] optimizes the spatial transformation of a fixed set of oriented Gaussian kernels over time to fit multiview video sequences. We build upon Dyn3DGS to extract dense correspondence of 3D Gaussians for deforming objects. Our key insight is that maintaining uniform Gaussian attributes improves modeling accuracy and physical consistency. Specifically, we keep the color, opacity, and scale of Gaussians constant during optimization, while allowing position and orientation change. We empirically found that enforcing color-consistent ellipsoids reduces optimization parameters and speeds up training, resulting in improved correspondence quality, especially when the video contains partial occlusion. Additionally, we only retain high-opacity Gaussians during training.

In line with Dyn3DGS, we use physics-inspired optimization objectives to guide optimization, ensuring physical plausibility and accurate long-term dense correspondence. The non-rigid physical modeling principles capture natural scene dynamics, enhancing the fidelity and stability of tracking over time. To overcome the challenges of our 4-view configuration, we strengthen the local rigidity and isometry objectives, improving the ability to handle limited observations.

To isolate the objects of interest, we use GroundingDINO [53] and Segment Anything [54] models to obtain masks for objects the robot interacts with. This allows us to filter out the interference of background objects, thus enhancing the optimization efficiency.

## 3.3 Graph-Based Neural Dynamics Learning

The dense tracking of object particles facilitates action-conditioned dynamics learning. Consider the scenario of an external robotic action applied to an object. The action can be represented as a sequence of end-effector positions $A = \{a_t\}_{0 \leq t \leq T}$. From the video sequence of the action, we can fit 3D Gaussians to the video using our modified Dyn3DGS, resulting in a collection of dynamic Gaussians: $X = \{X_t\}_{0 \leq t \leq T} = \{\mu_i^t\}_{1 \leq i \leq N, 0 \leq t \leq T}$, where $X_t$ denotes the set of Gaussians at time $t$, and $\mu_i^t$ denotes the position of a single Gaussian indexed $i$ at time $t$. The underlying physical dynamics function could then be depicted as

$$X_{t+1} = f(X_{0:t}, a_t) \tag{2}$$

for any timestep $t$. We approximate this dynamics function with a Graph Neural Network (GNN) architecture [55], parameterized by $\theta$. To construct the graph, we apply the farthest point sampling algorithm on the Gaussians $X$ to extract a sparse subset of control particles $\hat{X} = \{\hat{X}_t\}_{0 \leq t \leq T} = \{\hat{\mu}_i^t\}_{1 \leq i \leq n, 0 \leq t \leq T}$, with $n$ particles and pairwise distance threshold $d_v$, and use them as graph vertices. The robot end-effector position $a_t$ is also considered as a graph vertex. A bidirectional edge is connected if two vertices' distance is below a threshold $d_e$, resulting in the edge set $\hat{E}$.

The vertex encoder takes as input the motion of each particle $\hat{\mu}_i^t - \hat{\mu}_i^{t-1}$ in the previous $k$ time steps ($k = 3$ in our experiments) and a one-hot vector indicating whether the vertex belongs to the object or the robot end-effector. To model the relationship between the object and the table surface, the distance of every vertex to the surface is also included in the input. The edge encoder takes as input the 3-dimensional position difference of the two vertices and a one-hot vector indicating whether the edge is an object-object relation or an object-robot relation.

The graph encodes vertices and edges into latent features using shared vertex and edge encoders, performs message passing for $p$ timesteps ($p = 3$ in our experiments), and uses a shared decoder to

map vertex latent features back to a 3-dimensional output indicating the motion at time $t$. The entire process can be represented as

$$\hat{X}_{t+1,pred} = \hat{X}_t + f_\theta(\hat{X}_{t-k:t}, \hat{E}_t). \tag{3}$$

**Training.** The main loss function to train our dynamics network is the MSE prediction error:

$$\mathcal{L}_{pred} = \sum_{i=1}^{\tau} \|\hat{X}_{t+i,pred} - \hat{X}_{t+i}\|^2, \tag{4}$$

where $\tau$ is the look-forward horizon size indicating how many steps we recurrently perform future prediction. We take $\tau = 5$ in our experiments to get a balance between prediction stability and computational cost. During training, we randomly sample time steps $t$ from the training sequences to serve as the starting frame.

Empirically, we also find adding regularization terms to be helpful in some objects with complex shapes. These include an edge length regularization term penalizing the mean squared difference of edge lengths between subsequent frames, and a rigidity regularization term, applied to objects that are rigid or close to rigid, penalizing the mean squared difference between the predicted motions and a rigid transformation estimated from the predictions [56].

### 3.4 Dense Motion Prediction on 3D Gaussians

The flexibility of 3D Gaussians allows us to perform motion densification from the sparse subset of particle $\hat{X}$ to the entire collection of Gaussians $X$ while maintaining the ability to render photo-realistic videos. Taking an initial static collection of Gaussians $X = \{X_0\}$ and the action sequence $A = \{a_t\}_{0 \le t \le T}$ as input, we first perform recurrent future prediction on the subsampled graph vertices, outputting $\hat{X}_{0:T}$. Then, for each timestep $t$, we perform the following interpolation scheme to derive the center motions and rotations of each Gaussian:

**Graph Vertex Transformations.** In the first step, we need to calculate the 6-DoF transformation for each graph vertex. The outputs of the GNN are per-vertex motions, which directly serve as the 3D translations. For the 3D rotation, for each vertex $\hat{\mu}_i^t$, we solve for the rotation $R_i^t$ according to the motion of its neighbors from time $t$ to $t+1$:

$$R_i^t = \arg\min_{R \in SO(3)} \sum_{j \in \mathcal{N}(i)} \|R(\hat{\mu}_j^t - \hat{\mu}_i^t) - (\hat{\mu}_j^{t+1} - \hat{\mu}_i^{t+1})\|^2. \tag{5}$$

**Gaussian Transformations.** We adopt Linear Blend Skinning (LBS) [57, 58] to calculate the motions of Gaussian centers. Concretely,

$$\mu_i^{t+1} = \sum_{b=1}^{n} w_{ib}^t \left( R_b^t(\mu_i^t - \hat{\mu}_b^t) + \hat{\mu}_b^t + T_b^t \right), \quad q_i^{t+1} = \left( \sum_{b=1}^{n} w_{ib}^t r_b^t \right) \odot q_i^t, \tag{6}$$

where $w_{ib}^t$ is the weight associated between Gaussian $i$ and graph vertex $b$; $R_b^t, r_b^t$ are the matrix and quaternion representations of vertex $b$'s rotation at time $t$; $T_b^t$ is the translation of vertex $b$, which is directly predicted by our dynamics model. The blending weight between a Gaussian and a vertex is inversely proportional to their 3D distance:

$$w_{ib}^t = \frac{\|\mu_j^t - \hat{\mu}_b^t\|^{-1}}{\sum_{b=1}^{n} \|\mu_j^t - \hat{\mu}_b^t\|^{-1}}, \tag{7}$$

thus assigning larger weights to vertices that are spatially closer to the Gaussians. We start from the initial Gaussian centers and rotations at $t = 0$ and repeatedly calculate the blending weights and updated Gaussian centers and rotations, resulting in dense and smooth Gaussian motions over the entire video sequence.

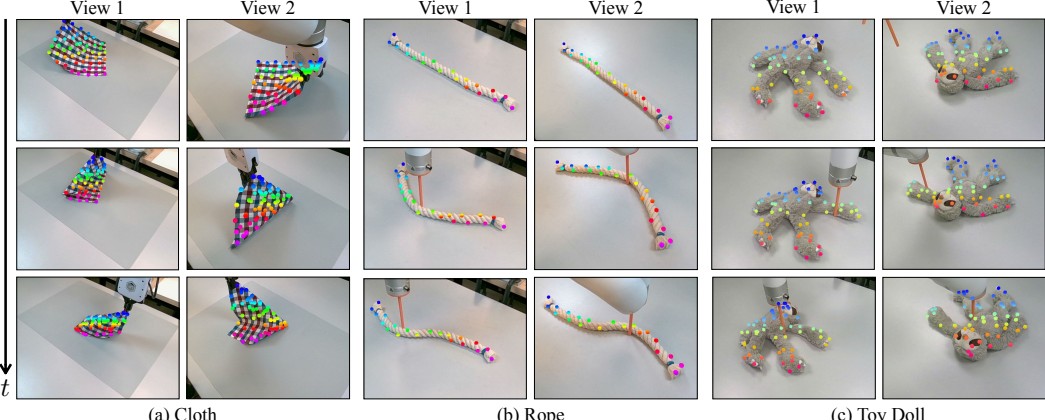

|  | View 1 | View 2 | View 1 | View 2 | View 1 | View 2 |
| --- | --- | --- | --- | --- | --- | --- |
| | (a) Cloth | | (b) Rope | | (c) Toy Doll | |

Figure 3: **Qualitative Results of 3D Gaussian Tracking.** We demonstrate point-level correspondence on the objects across various timesteps. Please check our website for more videos showcasing precise dense tracking even under different object deformations and occlusions.

### 3.5 Model-Based Planning

The dynamics network can also be used to perform model-based planning in robotics tasks. Given the multi-view RGB-D image of the object, we apply segmentation and point fusing to get the complete point cloud of the object. We then construct graph vertices and edges with farthest point sampling. Given a target configuration, we use a Model Predictive Control (MPC) [59] framework to plan for robot actions. The optimization objective is defined as the mean squared difference between the target state and the predicted object state if an action is applied. In our experiments, we apply the Model-Predictive Path Integral (MPPI) [60] trajectory optimization algorithm.

## 4 Experiments

### 4.1 Experiment Setup

We perform experiments on 3D tracking, action-conditioned video prediction, and model-based planning. To achieve this, we collect a real-world dataset, with a focus on acquiring multiview images synchronized with corresponding actions at each timestep. The dataset consists of 3 types of deformable objects: rope, cloth, and toy animals, totaling 8 object instances. For all objects, we capture RGBD images and record robot actions at 15 FPS. The depth is used to initialize the point cloud for Gaussian representation. Our data collection setup includes four cameras, strategically positioned at the corners of the workspace and oriented to provide a comprehensive top-down perspective. This setup allows us to collect synchronized multiview data necessary for our experiments.

### 4.2 3D Tracking with Dynamic 3D Gaussians

We initialize Gaussian tracking with a point cloud from four views at the first timestep and optimize our Dyn3DGS for dense correspondence. To evaluate our method, we compare it with advanced 2D tracking approaches by projecting 4-view results into 3D using our depth data and evaluate the best results. The unmodified Dyn3DGS is considered as a baseline to show our improvements within the 4-view real-world experiment setup. Following prior work [61], we evaluate median trajectory error (MTE) in millimeters, position accuracy ($\delta$) at thresholds of 2, 4, 8, and 16 millimeters (reporting the average), and survival rate with a threshold of 0.5 meters. For a fair comparison, we also project our 3D tracking results into 2D pixels and evaluate the same metrics, allowing comprehensive assessment in both 2D and 3D contexts.

The quantitative results in Tab. 1 demonstrate that our 3D tracking method outperforms all baselines, including SpatialTracker [15], which uses depth as a 3D prior. The results highlight our method's capability across various scenes and objects. Additionally, the qualitative results in Fig. 3 show that our Dynamic 3D Gaussian tracking provides precise dense correspondence, even under challenging conditions such as occlusions from robot and object deformations.

| Method | Metric | Rope | Cloth | Toy Animals | Metric | Rope | Cloth | Toy Animals |
|---|---|---|---|---|---|---|---|---|
| CoTracker [12] | 3D MTE [mm] ↓ | 46.96 | 55.84 | 51.84 | 2D MTE [mm] ↓ | 42.73 | 49.82 | 46.30 |
| PIPS++ [61] | | 49.37 | 58.10 | 58.45 | | 46.17 | 50.28 | 49.96 |
| SpatialTracker [15] | | 38.72 | 53.85 | 42.82 | | 32.29 | 46.26 | 37.49 |
| Dyn3DGS [8] | | 62.41 | 69.20 | 66.19 | | 57.39 | 61.28 | 60.17 |
| Ours | | **6.90** | **13.14** | **12.83** | | **4.92** | **11.72** | **10.94** |
| CoTracker [12] | 3D $\delta_{avg}$ ↑ | 79.28 | 75.79 | 75.46 | 2D $\delta_{avg}$ ↑ | 83.71 | 79.28 | 78.44 |
| PIPS++ [61] | | 69.83 | 66.92 | 68.95 | | 76.30 | 72.48 | 76.96 |
| SpatialTracker [15] | | 86.56 | 83.85 | 79.42 | | 92.36 | 90.52 | 89.62 |
| Dyn3DGS [8] | | 60.32 | 56.28 | 61.97 | | 67.20 | 62.28 | 67.96 |
| Ours | | **89.26** | **89.13** | **82.71** | | **93.27** | **92.18** | **94.19** |
| CoTracker [12] | 3D Survival ↑ | 94.41 | 95.19 | 92.26 | 2D Survival ↑ | 97.20 | **100** | 96.06 |
| PIPS++ [61] | | 92.61 | 91.39 | 85.73 | | 96.74 | 94.28 | 92.82 |
| SpatialTracker [15] | | **100** | 98.46 | 96.26 | | **100** | **100** | **100** |
| Dyn3DGS [8] | | 84.29 | 79.04 | 74.82 | | 87.83 | 82.14 | 79.32 |
| Ours | | **100** | **100** | **98.83** | | **100** | **100** | **100** |

Table 1: **Quantitative Results on Dynamic 3D Gaussian Tracking.** We labeled the ground truth for 200 frames per episode in 3D space for 1 or 2 object instances in each category, covering two episodes per object. Our Dyn3DGS-based tracking method outperforms all baselines, including the unmodified Dyn3DGS, in both 2D and 3D metrics.

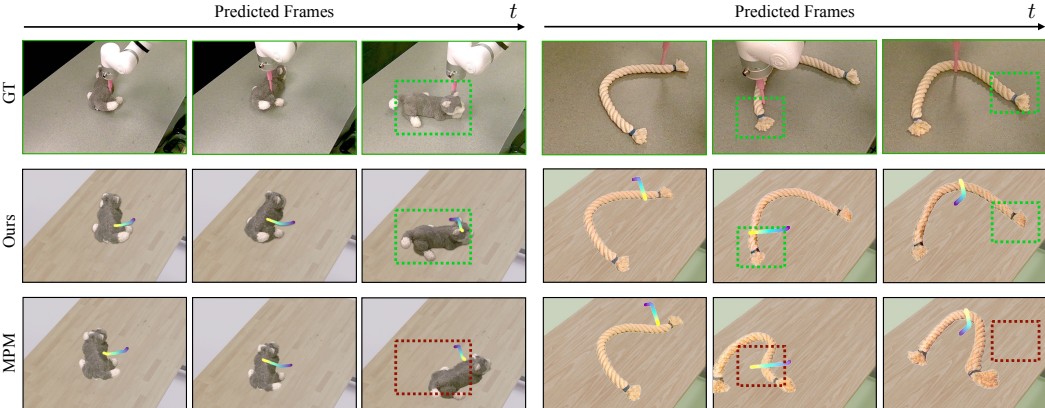

Figure 4: **Qualitative Results of Action-Conditioned 3D Video Prediction.** Our videos are generated by rendering predicted Gaussians on virtual backgrounds. Robot trajectories are visualized as curved lines (yellow: current end-effector positions, purple: history end-effector positions). Compared to the MPM baseline, our video prediction results align with the ground truth frames (GT) more accurately.

## 4.3 Action-Conditioned Video Prediction

Using dynamic 3D Gaussian reconstruction with tracking, we train the graph-based neural dynamics models with downsampled control particles. These particle motions are used to interpolate dense Gaussian kernel motion and rotations. To evaluate the quality of our dynamics prediction and 3D Gaussian rendering, we assess action-conditioned video prediction performance.

**Baselines.** Synthesizing action-conditioned object motions requires physics priors, and existing text-conditioned video prediction methods are not compatible with taking 3D robot actions as input. In this experiment, we consider 2 physics simulator-based baselines, *MPM* and *FleX*.

*MPM* is based on the Material Point Method simulation framework [62, 63]. Our baseline MPM uses the same simulation setting as previous works [22, 26], while adding support for two types of robot end-effectors: cylindrical pusher and gripper. For each object instance, we set the friction coefficient $\mu$ and Young's modulus $E$ (assumed uniform) to be two learnable parameters that are optimized from the training data using the CMA-ES algorithm. *FleX* is based on the NVIDIA FleX simulator. For all objects, we adopt the soft body simulator in FleX, reconstruct the initial object mesh from 3D Gaussians using alpha-shape mesh reconstruction. Similarly, we optimize the friction coefficient $\mu$ and stiffness coefficient $s$ (assumed uniform) of each object instance using CMA-ES.

**Metrics.** To evaluate the 3D video prediction performance, we adopt 3D distance metrics (3D Chamfer Distance, 3D Earth Mover's Distance), 2D segmentation similarity metrics ($\mathcal{J}\&\mathcal{F}$ score),

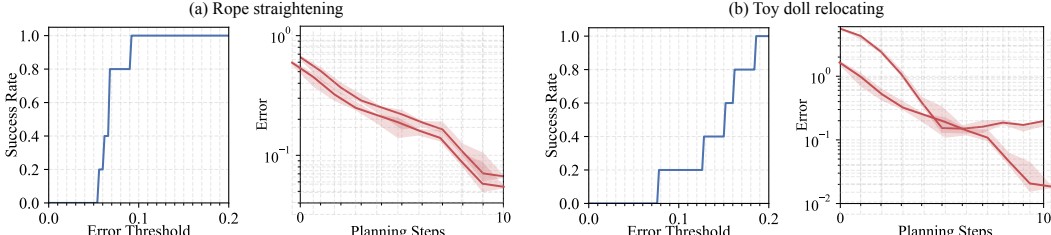

Figure 5: **Quantitative Results of model-based planning.** We perform each experiment 5 times and present the results as follows: (i) the median error curve relative to planning steps, with the area between 25 and 75 percentiles shaded, and (ii) the success rate curve relative to error thresholds.

and 2D image-based metrics (LPIPS). The metrics are complementary and together give a thorough comparison of future prediction accuracy and rendering quality.

**Results.** The results are listed in Tab. 2. Across all metrics, our method outperforms both baselines significantly, proving that we learn a more accurate dynamics model and give more realistic video prediction results. Our neural-based dynamics model effectively avoids the sim-to-real gap by learning from real-world data, thus improving the motion prediction accuracy under unseen robot actions and object configurations. This is also supported by the qualitative visualizations of predicted videos, shown in Fig. 4.

| Method | Metric | Cloth | Toy Animals | Rope | Metric | Cloth | Toy Animals | Rope |
|---|---|---|---|---|---|---|---|---|
| MPM | 3D Chamfer ↓ | 0.076 | 0.052 | 0.073 | $\mathcal{J}$ score↑ | 0.421 | 0.602 | 0.397 |
| FleX | | 0.136 | 0.069 | 0.074 | | 0.268 | 0.451 | 0.299 |
| Ours | | **0.064** | **0.030** | **0.053** | | **0.538** | **0.701** | **0.419** |
| MPM | 3D EMD ↓ | 0.097 | 0.053 | 0.066 | $\mathcal{F}$ score ↑ | 0.296 | 0.598 | 0.600 |
| FleX | | 0.126 | 0.066 | 0.068 | | 0.182 | 0.424 | 0.552 |
| Ours | | **0.067** | **0.035** | **0.048** | | **0.446** | **0.726** | **0.667** |
| MPM | LPIPS ↓ | 0.057 | 0.027 | 0.030 | $\mathcal{J}\&\mathcal{F}$ score ↑ | 0.359 | 0.600 | 0.498 |
| FleX | | N/A | N/A | N/A | | 0.225 | 0.438 | 0.426 |
| Ours | | **0.044** | **0.021** | **0.025** | | **0.492** | **0.714** | **0.543** |

Table 2: **Quantitative Results on Action-Conditioned 3D Video Prediction.** We evaluate a test sequence set for each object instance and present the results averaged by object category. Our method outperforms simulator baselines across all metrics.

### 4.4 Model-Based Planning

The action-conditioned dynamics model enables deployment in real-world robot planning tasks, demonstrating its generalizability to unseen object configurations. We choose rope straightening and toy animal relocating tasks to show our model's ability to manipulate highly deformable objects to target configurations. The quantitative results in Fig. 5 demonstrate that our approach effectively reduces error and achieves high success rates. The computational complexity and inefficiency of MPM and FleX baselines make them impractical to perform real-world planning tasks.

## 5 Conclusion and Limitation

In this paper, we introduce a novel approach for graph-based neural dynamics modeling from real-world data for 3D action-conditioned video prediction and planning. Our method reconstructs 3D Gaussians with cross-frame correspondences and learns a neural dynamics model that facilitates action-conditioned future prediction. The method outperforms baseline approaches in motion prediction and video prediction accuracy. We also showcase our planning performance for object manipulation tasks, highlighting our framework's effectiveness in real-world robotic applications.

**Limitation** Although our approach learns object dynamics directly from videos, collecting real-world data remains costly, and the perception module may fail when there are large occlusions or textureless objects. In future work, we aim to develop a more efficient data collection pipeline and a more robust perception system to reduce the learning cost and enhance the method's applicability to all kinds of real videos.

**Acknowledgments**

The Toyota Research Institute (TRI) partially supported this work. This article solely reflects the opinions and conclusions of its authors and not TRI or any other Toyota entity. We would like to thank Haozhe Chen, Binghao Huang, Hanxiao Jiang, Yixuan Wang and Ruihai Wu for their thoughtful discussions. We thank Binghao Huang and Hanxiao Jiang for their help on the real-world data collection framework design, and also thank Binghao Huang and Haozhe Chen for their help with visualizations.

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
