# OpenReview forum: "Dynamic 3D Gaussian Tracking for Graph-Based Neural Dynamics Modeling"
_robot-learning.org/CoRL/2024/Conference — CoRL 2024_

### Official Review · Reviewer_9VLH · 2024-07-11
**Combination of existing parts to an interesting and novel method**

**Originality:** 3
**Technical Quality:** 4
**Clarity Of Presentation:** 4
**Potential Impact:** 3
**Recommendation:** 3
**Confidence:** 5

**Review:**

### Strengths

**Novelty and potential**
Training an action-conditioned motion-model using 3D point tracking as supervision is novel and has great application potential for many different robotic applications.

**Evaluation**
The quantitative and qualitative evaluation is convincing and shows the efficacy of the proposed method. In particular, during the rebuttal, the authors presented further results on more complex scenes showing the performance of the method under the influence of shadows or occlusions.

**Rebuttal**
During the rebuttal phase, the authors were able to address most of my concerns regarding the paper with *extensive* answers and presented an impressive amount of additional results.

**Clarity**
The paper is overall well written, with easily comprehensible figures which support the general understanding.

### Weaknesses

**Methodological contribution**
While the presented method is overall novel, the novelty of the individual components—tracking and dynamics modelling—is limited, and the primary contribution comes from their combination. For me it seems that especially the tracking part of the method as described in 3.2 is just a slightly modified version of Dyn3DGS, with the key difference that the parameters of the Gaussians (except the mean) are kept constant over time and that the weights of the regularization losses are tuned for the method to function in a sparsely observed environment.

**Straightened start configuration**
The method shares the limitation of Dyn3DGS, that only parts of the scene can be tracked which are visible in the initial frame. This requires for example cloths to be in a straightened state in order to be properly modelled.

**Weaknesses resolved during rebuttal**
- Lacking a sufficient limitations section.
- Missing qualitative examples on complex scenes which contain shadows and occlusion.
- Model-based planning experiments without baselines. (Still no baselines, but sufficient justification why not)

**Quality Of The Limitations Section:**

3

**Questions For Rebuttal:**

**Issues**
- Evaluate tracking and video prediction on more complex scenes (shadows, more deformation, ...) *resolved*
- Add the unmodified Dyn3DGS as tracking baseline/ablation *resolved*
- Put the planning results into perspective by comparing to baselines *resolved*
- Extend limitations discussion *resolved*

**Questions**
- Since there is already a dependency on using depth cameras for the initialization of the 3D Gaussians, have the authors also considered using depth to guide the optimization of the tracking? *answered*
- What is used as the ground truth state for the calculation of the tracking metrics? *answered*
- Since the color of the 3D Gaussians is kept constant over time, wouldn't there be issues in the tracking if there were strong dynamic shadows, for example from the robot occluding the light source? *answered*
- How long does it take to train the dense tracking, especially in comparison to the baselines? It would be a strong limitation if the tracking took for example 10min to train, while the baselines were able to yield results after a few seconds. *answered*
- L149: "In the case where an external obstacle (for example, a table surface) is in the scene, the distance and direction of every vertex to the obstacle are also included in the input." Since this is not further explained in the paper: How are obstacles detected/modeled? *answered*

**Robotics Focus:**

4

**Summary Of Paper:**

The paper focuses on learning action-conditioned dynamics models from short sequences of RGB-D images that capture a robot interacting with a deformable object.  Initially, the authors use these observations for a vision-based 4D reconstruction of the object based on  Dyn3DGS. They then select a subset of the learned 3D Gaussians, which, combined with known robotic actions, are used to train a GNN-based dynamics model. The effectiveness of this method is demonstrated by evaluating the tracking performance of the 4D reconstruction and by assessing the dynamics model for planning and action-conditioned video prediction.

**Summary Of Recommendation:**

After the rebuttal I changed my recommendation to "weak accept". Although I am convinced of the novelty, potential and evaluation of the method, it has some shortcomings when it comes to a methodological contribution.

---

### Official Review · Reviewer_sKFF · 2024-07-19
**Interesting idea but lack of methodological contributions and convincing experiments.**

**Originality:** 2
**Technical Quality:** 2
**Clarity Of Presentation:** 3
**Potential Impact:** 3
**Recommendation:** 3
**Confidence:** 4

**Review:**

The paper proposes an interesting idea to learn 3D action-conditioned models of objects from real-world videos leveraging recent advances in point-tracking. Moreover, the real-world evaluation is extensive.

My primary concern relates to the novelty of the proposed dense tracking with Gaussian Splatting (GS). Neither the GS technique nor the physics-inspired regularization losses are a contribution of this paper, as presented also in the original paper [1]. However, the authors claim this to be “their proposed tracking method” (see lines 135-136). Moreover, the authors further claim that their  Dynamic 3D Gaussian tracking provides precise, dense correspondence, even under challenging conditions such as occlusions from the robot and object deformations. However, with multiple views, the robot occlusions are limited, and the deformations shown in the evaluation are not particularly strong. The paper would be much stronger if the method could account for larger deformations, including self-occlusions of the object.

Regarding the experiments, all the tracking baselines are single-view baselines. Did the author lift these baselines to multi-view? If not, I would suggest at least reporting the results from the best view or providing the results of the proposed method from only one view. Moreover, Dynamic 3D Gaussians[1] is missing.

The authors put some emphasis on the action-conditioned video prediction, claiming that compared to related work on video prediction for robotics, their work “facilitates dynamics learning to support more complex robotic manipulation tasks.”. However, this statement is not supported by experimental results. I wonder why the authors do not compare with other data-driven approaches, such as visual foresight [2], but instead claim that “existing text-conditioned video prediction methods are not compatible with taking 3D robot actions as input.”

My final comment relates to Figure 4. In the rope experiments, I wonder how it is possible that in the second frame of the MPM simulation, the pusher moves the wrong side of the rope (which in the previous frame is far from the starting position of the push). This looks to me like either the wrong frame or an implementation error.

[1] Luiten, J., Kopanas, G., Leibe, B. and Ramanan, D., 2023. Dynamic 3d Gaussians: Tracking by persistent dynamic view synthesis. arXiv preprint arXiv:2308.09713.

[2] Finn, C. and Levine, S., 2017, May. Deep visual foresight for planning robot motion. In 2017 IEEE International Conference on Robotics and Automation (ICRA) (pp. 2786-2793). IEEE.

### Clarity
The paper is clear and easy to follow. The only section that I believe requires some revision is related to the Gaussian transformations in section 3.4. Equation (9) should be revised. Should the term $\mu_j^t$ be $\mu_i^t$? What is $T_b^t$? Moreover, it is unclear how the blending weights are calculated. It is mentioned that it is inversely proportional to the 3D distance of the Gaussian i and graph vertex b but more details could be provided in order to ensure replicability.

### Strengths
* Combining 3D tracking methods to learn object dynamics from real-world objects is a novel application.
* The authors provide extensive real-world experiments.
* The paper is well-written and easy to follow.


### Weaknesses
* I challenge the novelty of the tracking approach proposed in this work.
* The claimed benefits of the Dynamic 3D Gaussian tracking under challenging conditions (occlusions and deformations) are not convincingly supported. The evaluations do not show strong deformations, and multiple views minimize robot occlusions.
* Some relevant baselines are missing in the current evaluation.
* Minor methodological contribution.

**Quality Of The Limitations Section:**

3

**Questions For Rebuttal:**

* Would the learned model generalize to unseen objects?
* Clarify how the tracking method differs from Dyn3DGS and, if it is different, integrate it as a baseline.
* Clarify how the tracking results with respect to monocular baselines are conducted.
* Clarify Figure 4.
* Support the statement that your method “facilitates dynamics learning to support more complex robotic manipulation tasks.” by comparing it with other learned dynamics models (e.g., Visual Foresight).
* MINOR: Can you expand on the computational complexity of the overall method and the planning? The videos showed that it required 20x, so I wonder where the computational bottleneck is.
* MINOR: Revise equation 9.
* MINOR: Expand on the computation of the blending weights.
* CURIOSITY: How do you label ground truths for real-world evaluation? This is quite a challenging task, especially under deformations. It would be of great value to share the labeling approach.

**Robotics Focus:**

4

**Summary Of Paper:**

The authors propose a novel approach for learning 3D object dynamics from real-world videos of robot interactions. The method involves generating ground-truth object states with Gaussian Splatting (GS) and learning a Graph Neural Network (GNN) from a set of control points of the labeled data. The pre-trained GNN can then be used to predict 3D object states conditioned on robot actions and render future object states to achieve action-conditioned video prediction.

**Summary Of Recommendation:**

The proposed idea is interesting and relevant for robotics, but I challenge the real contribution of the work. Moreover, some claims of the authors are not well supported by the experiments.

---

### Official Review · Reviewer_tMai · 2024-07-22
**Overall, I think it's a good paper that presents a technically sound approach for training particle-based dynamics models with real-world videos.**

**Originality:** 3
**Technical Quality:** 4
**Clarity Of Presentation:** 3
**Potential Impact:** 3
**Recommendation:** 3
**Confidence:** 5

**Review:**

Strengths
1. I like the overall direction very much. Combining Gaussian Splatting (GS) with a particle-based dynamics model is a natural way to enable dynamics model training without using dense particle trajectory labels, which are hard to obtain in real-world videos. The proposed approach to achieve this goal is technically sound. Additionallly, to my knowledge, this is the first work to present a framework that combines GS and dynamics models in the context of robot manipulation.
2. The paper is easy to follow, although minor edits are needed to correct a few typos and clarify some points.
3. The effectiveness of the proposed method is demonstrated across point tracking, video prediction, and model-based planning.

Weaknesses
1. Novel contributions aren't clearly presented in Section 3.2. I believe most of Sec. 3.2., including equations (1)-(4), is from prior work [17], so I am not convinced that this much space is necessary to present Sec. 3.2. The difference between [17] and this paper in enabling dense point tracking needs to be clarified. For the same reason, Table 1 might be moved to the supplementary material if there wasn't much change between [17] and this paper in dense point tracking, as Table 1 would just show the results of [17] on the new datasets in that case. If there were any design changes between [17] and this paper, I encourage the authors to add ablation results in Table 1 to show any performance difference made by new changes.

2. It would be beneficial to include more clarifications in the paper regarding the following potential limitations:
- I believe the dynamics model cannot be generalized over object categories, meaning the dynamics model needs to be trained for each object category separately.
- The objects used during training and testing appear to be the same, so it is unclear whether the method can generalize to novel object instances within the same category (i.e., new toy dolls that the dynamics model hasn't seen during training).

3  Typos or missing details.
- In line 100, what is J?
- Inconsistent point neighborhood notations in Eq. (2)-(4), and (8): knn(i,k), knn_{i,k}, N(i)
- what is T^t_b in Eq (9)?
- Mu^t_i instead of Mu^t_j in Eq (9)?
- In line 188, what does "with FPS" mean?

**Quality Of The Limitations Section:**

2

**Questions For Rebuttal:**

I would like to see answers to the questions I raised in points 1 and 2 of the Weaknesses section above.

**Robotics Focus:**

4

**Summary Of Paper:**

The paper proposes a new framework that combines a particle-based dynamics model with Gaussian splatting, enabling the training of dynamics models using unlabeled multi-view (posed) video observations. The authors evaluated the proposed model on real-world videos of a robot arm interacting with deformable objects, demonstrating strong results in 3D point tracking, video prediction, and model-based planning.

**Summary Of Recommendation:**

Although the paper needs some polishing to fix a few typos and clarify the points I raised in my review, overall, I think it's a good paper with strong results. It successfully demonstrates the use of particle-based dynamics models trained with rendering supervision for robot manipulation. After rebuttal: I thank the authors for all their significant efforts in addressing the reviewers' concerns. I am happy with the answers provided by the authors, so I will keep my original "weak accept" rating.

---

### Author Rebuttal · Authors · 2024-08-12

Our revised main paper and supplement are in the rebuttal file.

---

### Decision · Program_Chairs · 2024-09-04

**Decision:**

Accept

**Comment:**

Thank you for your submission to CoRL 2024. The reviewers liked many aspects of the paper and felt it was a promising line of research. They specifically pointed out that they liked the direction of combining Gaussian splatting and particle-base dynamics, the well-written paper, the effectiveness of the method, and the real-world learning and experiments.

The reviewers raised concerns about unclear points of novelty, wanting clarification of limitations, concerns about the experiments (e.g., fairness and single-view vs multiview), a few unsupported claims, and more methodological contributions.

The authors' thorough responses to the initial reviews were compelling and appreciated by the reviewers and the AC. While there are lingering concerns (which we encourage the authors to address for camera-ready), they found that the response was sufficient to raise the recommendation.